# Diversity and Molecular Evolution of Odorant Receptor in Hemipteran Insects

**DOI:** 10.3390/insects13020214

**Published:** 2022-02-21

**Authors:** Jiahui Tian, Youssef Dewer, Haoyuan Hu, Fengqi Li, Shiyong Yang, Chen Luo

**Affiliations:** 1School of Ecology and Environment, Anhui Normal University, Wuhu 241002, China; tianjh0920@163.com (J.T.); haoyuanhu@126.com (H.H.); 2Beijing Key Laboratory of Environment Friendly Management on Fruit Diseases and Pests in North China, Institute of Plant and Environment Protection, Beijing Academy of Agriculture and Forestry Sciences, Beijing 100097, China; luochen1010@126.com; 3Central Agricultural Pesticide Laboratory, Agricultural Research Center, Phytotoxicity Research Department, Dokki, Giza 12618, Egypt; dewer72@yahoo.com; 4Collaborative Innovation Center of Recovery and Reconstruction of Degraded Ecosystem in Wanjiang Basin Co-Founded by Anhui Province and Ministry of Education, Anhui Normal University, Wuhu 241000, China

**Keywords:** hemiptera, odorant receptor, phylogeny, molecular evolution

## Abstract

**Simple Summary:**

Insects’ behavior and ecology are closely related to their chemosensory systems, during which odorant receptors (ORs) play an essential role in host recognition. Although OR gene evolution has been studied in many insect orders, a comprehensive evolutionary analysis and expression of OR gene gain and loss events among diverse hemipteran species are still needed. In this study, we identified and analyzed the OR genes from hemipteran species systematically. The number of OR genes discovered in each species ranged from less than ten to hundreds. Gene gain and loss events of OR have occurred in several species in the seven major clades classified through phylogenetic analysis. Then, we discovered the amino acid differences between species to understand the molecular evolution of OR in the order Hemiptera through positive selection. This study lays a foundation for subsequent investigations into the molecular mechanisms of Hemiptera olfactory receptors involved in host recognition.

**Abstract:**

Olfaction is a critical physiologic process for insects to interact with the environment, especially plant-emitted volatiles, during which odorant receptors (ORs) play an essential role in host recognition. Although OR gene evolution has been studied in many insect orders, a comprehensive evolutionary analysis and expression of OR gene gain and loss events among diverse hemipteran species are still required. In this study, we identified and analyzed 887 OR genes from 11 hemipteran species. The number of OR genes discovered in each species ranged from less than ten to hundreds. Phylogenetic analysis revealed that all identified Hemiptera OR genes were classified into seven major clades. Gene gain and loss events of OR have occurred in several species. Then, by positive selection, we discovered the amino acid differences between species to understand the molecular evolution of OR in the order Hemiptera. Additionally, we discussed how evolutionary analysis can aid the study of insect–plant communication. This study lays a foundation for subsequent investigations into the molecular mechanisms of Hemiptera olfactory receptors involved in host recognition.

## 1. Introduction

Insect behavior and ecology are closely related to their chemosensory system and are involved in various activities, such as host plant selection, mating, intra-specific communication, and avoidance behaviors [1,2]. A key step in insect chemosensation is the detection of chemicals by receptor proteins located on the membranes of peripheral sensory neurons, which convert chemical signals into electrical signals that travel into the insect’s central nervous system, ultimately resulting in behavioral responses. Chemoreceptors primarily include three main receptor families [3], gustatory receptors (GRs), ionotropic receptors (IRs), and odorant receptors (ORs). GRs contribute to detecting carbon dioxide, sugars, bitter components, salts, and several gustatory pheromones [3,4,5,6]. IRs mainly detect acids and amines [7], and ORs are key receptors involved in the detection of volatile compounds, including host plant volatiles [6].

The insect odorant receptors are seven-transmembrane domain proteins [8] that are classified into an odorant receptor co-receptor (Orco) and conventional ORx. The Orco and ORx form a heterotetrameric receptor complex on the membrane of olfactory sensory neurons, which is involved in olfactory recognition [9,10]. Orco is widely expressed and highly conserved in insects, highlighting its critical role in olfaction [11]. The conventional ORx varies in insects [12]. ORs have been discovered in various orders of insects, including Diptera, Hymenoptera, Lepidoptera, Coleoptera, and Hemiptera since the discovery of the first insect OR in *Drosophila melanogaster* [13,14] by using the whole-genome sequencing method. Additionally, the number of ORs in different species varies greatly [15]. For example, *Drosophila melanogaster* has 62 ORs [14] while *Tribolium castaneum* has 259 ORs [16], indicating a variable evolution of impact OR genes [17]. In Coleoptera, the diversity of OR genes in a given species is correlated with its ecology, and the number of ORs is also parallel to the host [15].

The evolution of ORs between different insect species remains unclear. This is partly due to the lack of genomic annotation information in some species, especially non-model species. Although some species have transcriptome data, OR genes have high and rapid diversity and low expression levels in different tissues, making OR annotation incomplete [18]. With the increase in genome annotation data, some gene families and genera have been annotated and compared successively in Diptera, Lepidoptera, Hymenoptera, and Coleoptera [15,19,20,21]. According to these findings, the OR gene family has been gained and lost numerous times in insects and has undergone rapid evolution, associated with host specificity and insect sociality [21,22]. In Hemiptera, although ORs have been identified and annotated in several species [12,23], the gain, loss, and evolution of OR genes among different species have not been well-explained.

In Hemiptera, the types and numbers of hosts in different species vary significantly. In the Delphacidae family, for example, *Nilaparvata lugens* is a monophagous pest that only feeds on rice, whereas *Laodelphax striatellus* is an oligophagous insect, which mainly feeds on rice but also feeds on other gramineous plants, including corn and tares [24]. The *Aphis gossypii* and the *Bemisia tabaci* in the Aphididae family are polyphagous insects. They host over 700 plant species globally [25,26]. The Hemiptera includes insects that feed on vertebrate blood, such as *Rhodnius prolixus*. It also includes *Cimex lectularius*, which feeds on human blood [12,27]. In the Hemiptera with such complex hosts, how the olfactory receptor adapts to different hosts has not been well-explained.

It is unclear whether the number of olfactory receptors varies in hemipteran species or whether there is any gene loss or gain as in other insect orders and adaptive evolution. Therefore, this study referred to the bioinformatics pipeline to systematically annotate the OR of 11 species from nine Hemiptera families, one species from Thysanoptera, and one from Corrodentia, closest to Hemiptera. We compared the sequence diversity and evolution of ORs in different species through phylogenetic and selection pressure analysis. Results of the present study may lay a foundation for further study of the function of Ors.

## 2. Materials and Methods

### 2.1. Genome Data and Integrity Assessment

For hemipteran insects, we selected 11 species (*Nilaparvata lugens*, *Halyomorpha halys*, *Cimex lectularius*, *Apolygus lucorum*, *Gerris buenoi*, *Rhodnius prolixus*, *Diaphorina citri*, *Acyrthosiphon pisum*, *Aphis gossypii*, *Trialeurodes vaporariorum*, and *Bemisia tabaci*), of which three cryptic species (Middle East-Asia Minor 1 (MEAM1), Mediterranean (MED), and Sub-Saharan Africa-East and Central Africa (SSA-ECA) were selected by *B. tabaci*. Furthermore, we selected *Pediculus humanus* of Corrodentia and *Frankliniella occidentalis* of Thysanoptera as outgroups of Hemiptera. Appendix A shows the genome assemblies of these species.

We assessed the genomics of each species using BUSCO v5.1.2 [28]. In *F. occidentalis* and *P. humanus*, we used the insecta_odb10 database (https://busco-data.ezlab.org/v4/data/lineages/insecta_odb10.2020-09-10.tar.gz) (accessed on 8 February 2022), and in the hemipteran insects, we used the hemiptera_odb10 database (https://busco-data.ezlab.org/v4/data/lineages/hemiptera_odb10.2020-08-05.tar.gz) (accessed on 8 February 2022).

### 2.2. Species Tree

The genome-annotated proteins of each species were processed and filtered using OrthoFinder (v.2.3.8) [29] to provide information about orthologous gene families. Orthoclades in OrthoFinder are defined as homologous genes of a single gene from the last universal common ancestor of the species under examination. The 1083 orthoclades were used to construct the species tree with a minimum of 83.3% of species having single-copy genes in any orthoclade [30,31].

### 2.3. Identification and Annotation of Odorant Receptor Family

We referred to a bioinformatics pipeline [32,33] to identify and annotate the members of the ORs in Hemiptera. We used the protein sequences of the known gene family in insects, downloaded from the National Center for Biotechnology Information (NCBI) as a database [12,23,25,34,35,36,37,38,39,40], and the database was used to construct a hidden Markov Model (HMM) profile for the family. The genome sequence, proteome sequence, and general feature format (GFF) annotation file of each identified hemipteran species were used as input files.

For the specific process, we first launched BLASTP and HMMER to search for OR gene families already present in the input GFF file. We filtered BLAST and HMMER hits with a default cut-off E-value of 1× 10^−5^. The screen and retain sequences covered two-thirds of the query sequences or over 80% of the complete proteins. Results of the two searches were pooled to generate a new GFF file. Then, we used TBLASTN to search for unannotated regions of protein homologs in the genome sequence. The gene model mapper (GeMoMa) tools based on homology and intron position conservation were employed to develop new gene models. Additionally, we used the input HMM configuration file to check the existence of gene family-specific protein domains. Subsequently, we utilized all of the proteins obtained in the first two steps as input data to perform a second search. NR annotation was applied to all of these putative OR genes, and all of the non-ORs were excluded from analysis. Finally, OR genes were finally identified using their specific seven transmembrane domains.

We utilized MAFFT v.7.037b [41], IQ-TREE [42] to construct the maximum likelihood tree of OR genes of each species and NCBI search to verify the accuracy of OR genes with the obtained non-redundant data set. OR genes were nominated according to the position of OR genes in the phylogenetic tree of each species.

### 2.4. Phylogenetic Analysis of OR Families

We aligned the identified protein sequences of the OR family in all species utilizing MAFFT (“-auto” option) [41]. IQ-TREE (v.2.1.2) was employed to estimate the best fit substitution models (best-fit model: LG + I + G4 chosen according to BIC) and gene family trees [42]. Additionally, node support was estimated with 1000 bootstrap replicates [43]. We created the tree using FigTree (v.1.3.1) and rooted it in the Orco lineage. 

### 2.5. Molecular Evolutionary Analysis

Genes subjected to positive selection were analyzed using the branch-site model in the codeml program in PAML v.4.9 software package [44]. Then, we aligned the nucleotide sequences using ClustalW (codons) [45] and then submitted the maximum likelihood tree and multiple sequence alignment files to the PAML software to calculate the ratio of dN (non-synonymous substitution rate) to dS (synonymous substitution rate) at each site. The magnitude of dN/dS value (ω) represents the type of selection: ω < 1 for negative selection, ω = 1 for neutral selection and ω > 1 for positive selection [46]. We employed the branch and branch-site models to check the positive selection of the branch, where the Aleyrodidae is located in clades two and four, and the branch where *H. halys* is located in clades three and four. In the branch model, the hypothesis was that the two-ratio model was significantly different from the one-ratio model (null, M0 one-ratio model), where only one ω value was estimated across the phylogeny. We performed the likelihood ratio test to check positive selection in the branch-site model. Finally, we mapped the amino acids under positive selection onto the odorant receptor topology to examine the distribution of the inferred positively selected sites using TMHMM.

## 3. Results

### 3.1. The Number of OR Genes in Different Species of Hemiptera Varied 

We used BUSCO to evaluate the integrity of 13 species downloaded from Insect Base, NCBI, Ensembl, and whitefly genome databases, among which *B. tabaci* includes the genomes of three cryptic species. Except for the whole BUSCO genes of *D. citri*, which was 88%, the complete BUSCO genes of the other selected species were above 90% (Figure 1). This result verifies the integrity of the genome used for subsequent identification of ORs.

We annotated 887 OR genes from the genomes of 11 hemipteran species (including three cryptic species in *B. tabaci*) (Figure 2, Appendix A). The genetic relationship between species in the species tree was consistent with a previous report [47]. The number of OR genes in different species ranges from 9 to 155 (Figure 2). We identified 11 and 39 OR genes in the genomes of *P. humanu**s* and *F. occidentalis*, respectively. Additionally, we identified 54, 152, and 155 OR genes in genomes of the herbivorous hemipteran insects *N. lugens*, *H. halys*, and *A. lucorum*, respectively, and 102,100 and 47 OR genes in *D. citri* in Chermidae, *A. pisum*, and *A. gossypii* in Aphididae, respectively. In Aleyrodidae, we identified 20 OR genes in the genome of *T. vaporariorum* and 13, 12, and 9 OR genes in genomes of the whitefly SSA-ECA, MEAM1, and MED, respectively. In genomes of the blood-sucking hemipteran insects *R. prolixus* and *C. lectularius*, 129 and 63 OR genes were identified, respectively. In the genome of *G. buenoi* that feeds on body fluids of small worms and dead fish, 30 OR genes were identified. Based on the number of exons and introns in the identified sequences, we found that the number of exons in the complete OR sequence varied from 5 to 9 (Appendix A). Protein sequences, nucleotide sequences, and the GFF file of all ORs used in this study are available in Appendix A.

### 3.2. OR Genes in Different Species Have Specific Expansion and Contraction

Seven major subfamilies of ORs that contain the ORs of Hemiptera were detected in our analysis (Figure 2). The Orco genes of all species are clustered together due to their highly conserved amino acid sequences and are located at the base of the entire phylogenetic tree, forming the root of the phylogenetic tree. Except for clades three and six, which had less than 50% support, other clades received stronger support (Figure 3 and Appendix A).

The OR genes among the seven subfamilies showed systemic gain and loss. Clade One contained only six ORs, each of which comes from *N. lugens*, *H. halys*, *A. lucorum*, *R. prolixus*, *C. lectularius,* and *G. buenoi*. Clade Two contained 43 OR genes that were separated into two main branches, one of which had 20 ORs and 19 of which were from the Aleyrodidae. Another branch had 23 OR genes and 14 of them were from *C. lectularius*. Clade Three contained 163 ORs, which were separated into two main branches. One branch contained 46 OR genes and 41 were from *H. halys*; the other branch contained 117 OR genes, and all were from *A. pisum* and *A. gossypii*. Clade Four contained 88 ORs, which were separated into three branches: the first branch had 36 OR genes and 19 of them were from the Aleyrodidae; the second branch had 20 OR genes and 12 of them were from *A. lucorum*; the third branch had 32 OR genes and 25 of them were from *H. halys*. Clade Five had 144 ORs, including a branch with *R. prolixus*-specific expansion. Clade Six had 115 ORs, and 91 of them were *D. citri*, accounting for 89% of the ORs identified in this clade. Clade Seven had 315 ORs, including a branch with *N. lugens*-specific expansion and two branches with *A. lucorum*-specific expansion.

### 3.3. OR Genes in Aleyrodidae and Halyomorpha Halys Undergo Positive Selection

The number of ORs in *H. halys* is 152, with obvious gene expansion. In contrast, the ORs genes in Aleyrodidae had a specific contraction. Therefore, the OR genes of *H. halys* and Aleyrodidae were selected for selection pressure analysis. The branch and branch-site models of the CODEML program [44] in the PAML v4.9 package are the most commonly used models for selection pressure analysis. We selected two branches of the Aleyrodidae OR gene in Clades two and four and two branches of *H. halys* OR gene in Clades three and four for labeling (Figure 3). For the branch model, likelihood ratio tests revealed that log-likelihood values under the free ratio model (M1) were significantly higher (*p* < 0.001) than those under the one ratio model (M0) when branches one, two, three, and four were selected as foreground branches (Table 1). These results indicated that selective pressure varies among branches in each targeted clade and that some of these proteins have evolved under positive selection. 

In the branch-site model, we failed to detect any site with a posterior probability greater than 0.95 when Branches One, Three, and Four were selected as foreground branches (Table 2). In contrast, we detected nine sites (90 G, 206 I, 222 E, 271 V, 699 C, 711 A, 761 G, 775 G, and 792 E) that had undergone positive selection, and the posterior probability values were greater than 95% only when Branch Two was selected as the foreground branch (Table 2). This means that positive selection sites can only be identified in Branch Two. For ORs, the TMHMM algorithm predicted the NH2 tail intracellular direction and the COOH tail extracellular direction, which supports the “reverse” membrane topology of insect chemoreceptors [8,48,49]. We mapped all assumed positive selection sites in Branch Two onto the topology of OR. When PP > 95%, we observed that three of the nine positive selection sites located outside the transmembrane region (ER), four of the nine positive selection sites inside the transmembrane region (IR) and two sites in the sixth transmembrane region of the OR gene (Figure 4).

## 4. Discussion

We evaluated the relative effectiveness of our method for OR gene detection in genome. We evaluated the relative effectiveness of our method for detecting OR gene pools in the genome. By comparing the number of OR genes retrieved in this study with those found in previous studies in Hemiptera, *H. halys* [39], *R. prolixus* [12], *A. lucorum* [17], *A. pisum* [23], *A. gossypii* [25], and *N. lugens* [50], we detected more OR genes than previously reported (Figure 2). The use of antenna transcriptome sequences in previous studies contributed to an increasing number of ORs in *H. halys* [39]. Among the species reported using genome identification, the most recent genome version has much higher quality than the older versions used for OR identification. Furthermore, the applicability of the bioinformatics pipeline and its advantages in annotating gene families in genomes of varying quality have been proven in nine species to identify chemosensory receptor genes, including the model organism *Drosophila melanogaster* [32,51,52]. The results obtained through the bioinformatics pipeline were more accurate than those annotated by augustus-ppx. In addition, to overcome putative gene model errors, the pipeline method implements some filtering steps to determine the correctness of predicted coding sequences. For example, hmmer searches were performed to identify protein family domains in all newly annotated sequences, resulting in more complete and sufficient identification of OR genes [32]. All of these indicate that this process is reliable and feasible in the identification of insect OR genes.

The number of OR genes in different Hemipteran species has obvious gain and loss events. Gene family expansion and contraction are considered the outcome of random genomic drift or environmental adaptation [53,54]. Studies have shown that the number of OR has an association with the breadth of its host. For example, in Coleoptera, the stenophagous *Agrilus planipennis* has fewer ORs than the oligophagous *Leptinotarsa decemlineata* and *Dendroctonus ponderosae*, which have smaller complements of Ors than the polyphagous or scavenging species *Anoplophora glabripennis*, *Onthophagus taurus*, and *Tribolium castaneum* [22]. However, the number of OR genes in Hemiptera is not parallel to that of host breadth. The biotope of different species differs considerably. Some species live on land, while others perched on static water, such as *G. buenoi*. In terms of feeding habits, some are herbivorous insects, some feed on blood (*R. prolixus*), and others feed on small insects (*G. buenoi*). Among the herbivorous species, there are monophagous (*N. lugens*), oligophagous (*L. striatellus*), and polyphagous (*B. tabaci*) species. All of these factors may be linked to the differences in ORs number. Although both *R. prolixus* and *C. lectularius* feed on vertebrate blood [12,27], *R. prolixus* ORs are twice as numerous as *C. lectularius.* In the herbivorous Hemiptera, both *H. halys* and *B. tabaci* are polyphagous [55,56]. A total of 152 OR genes were identified in *H. halys*, while only 9, 12, and 13 were identified among the three cryptic species of *B. tabaci*, respectively. It demonstrates that in hemipteran insects, the number of OR genes in the species is not directly associated with the type and number of hosts. 

Some orthologous genes showing high sequence homology may have similar spatial structures and biological functions [20]. In Hemiptera, the functional studies of OR receptors are relatively less. However, previous studies have characterized the function of OR genes in some species of Hemiptera (Appendix A). In *A. pisum*, Apis_OR17 and Apis_OR28 mainly identify plant volatiles [57,58], and Apis_OR6 is mainly linked to the alarm pheromone EBF and geranyl acetate [59]. Among them, Apis_OR28 was observed in Clade Three, where the OR gene specifically expanded into *A. pisum*. Apis_OR6 and Apis_OR17 were mainly observed in Clade Seven. In *A. lucorum*, Aluc_OR29 and Aluc_OR39 mainly recognize compounds that attract females [60,61], and Aluc_OR77 as a sex pheromone receptor mainly recognizes (E)-2-hexenyl butyrate and hexyl butyrate [17]. Aluc_OR39 was observed in Clade Three, Aluc_OR29 in Clade Six, and the sex pheromone receptor Aluc_OR77 in Clade Seven. In *C. lectularius* [62], Clec_OR26 mainly recognizes the aggregation pheromone, and Clec_OR16 and Clec_OR49 mainly recognize human volatiles for host location; Clec_OR28, Clec_OR34, and Clec_OR37 mainly recognize the repellent compounds terpenes and terpenoids. We observed Clec_OR16, Clec_OR34, and Clec_OR37 in Clade Four; Clec_OR28 in Clade Five; and Clec_OR26 and Clec_OR49 in Clade Seven. A putative inference is that the types of ligands that additional OR genes with high homology may bind by combining the function of OR genes in Hemiptera with their positions in the phylogenetic tree. Additionally, through phylogenetic analysis, it is speculated that some new genes in some species have similar functions to known genes, which will provide a reference for subsequent functional studies.

Interestingly, the specific expansion of OR genes may be associated with the existence of positive selection, and the acquisition of new hosts may be driven by positive selection [20]. *H. halys* has specific expansion into Clades Three and Four. We discovered that these two branches were being driven by positive selection, and nine positive selection sites were identified in the OR gene of *H. halys* in Clade Three. Two of the sites with PP > 95% are located at the sixth transmembrane position of OR, and the amino acids of this transmembrane region are considered to be highly conserved [10]. Additionally, the specific expansion of the OR gene in *A. pisum* undergoes a positive selection [23]. This finding indicates that gene duplication is necessary for hemipteran insects to acquire new chemosensory functions and is driven by changes in ligands [63]. For example, lineage-specific duplication of OR genes contributes to adaptation to specific ecological conditions in Drosophila [64]. As we all know, hemipteran insects rely mainly on their chemosensory system to recognize chemicals in the ambient environment, and OR genes under positive selection pressure may drive the acquisition of new hosts. This may also explain why *H. halys* and *B. tabaci* have a wide host range. Additionally, this provides valuable information for the various odors and tastants encountered by hemipteran insects when they explore new habitats and environments.

## 5. Conclusions

In summary, we identified the OR gene library and found seven major subfamilies in 887 OR genes in the 11 species of Hemiptera (*B. tabaci* contains three cryptic species). We discovered that OR gene has a specific expansion in *N. lugens*, *H. halys*, *A. lucorum*, *D. citri*, *A. pisum*, and *A. gossypii*. In Aleyrodidae, the OR gene has obvious gene loss. The expansion process of the OR gene in *H. halys* has undergone positive selection, resulting in the discovery of candidate genes involved in olfactory recognition under positive selection pressure. Results of the present study may lay the foundation for future investigation into the function of ORs in hemipteran insects.

## Figures and Tables

**Figure 1 insects-13-00214-f001:**
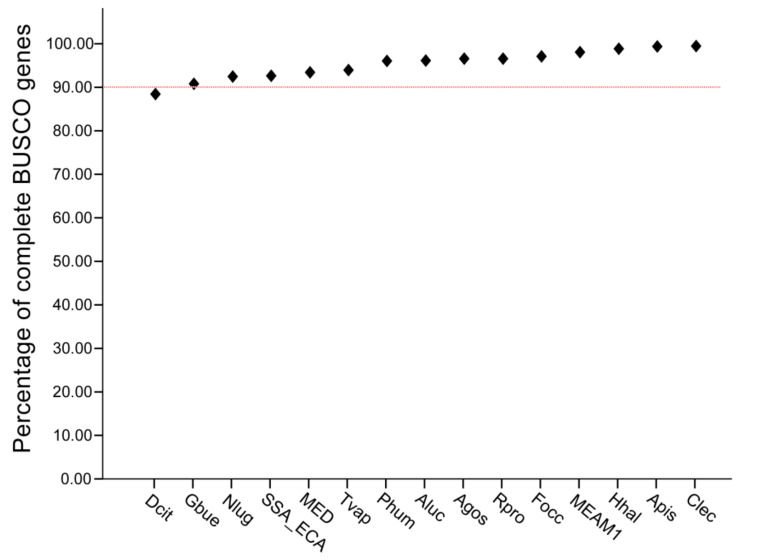
The completeness threshold (% of complete BUSCO genes) of the species genome for OR gene identification. *Pediculus humanus* (Phum), *Frankliniella occidentalis* (Focc), *Nilaparvata lugens* (Nlug), *Halyomorpha halys* (Hhal), *Cimex lectularius* (Clec), *Apolygus lucorum* (Aluc), *Rhodnius prolixus* (Rpro), *Gerris buenoi* (Gbue), *Diaphorina citri* (Dcit), *Acyrthosiphon pisum* (Apis), *Aphis gossypii* (Agos), *Trialeurodes vaporariorum* (Tvap), and *Bemisia tabaci* (MEAM1, MED, and SSA-ECA).

**Figure 2 insects-13-00214-f002:**
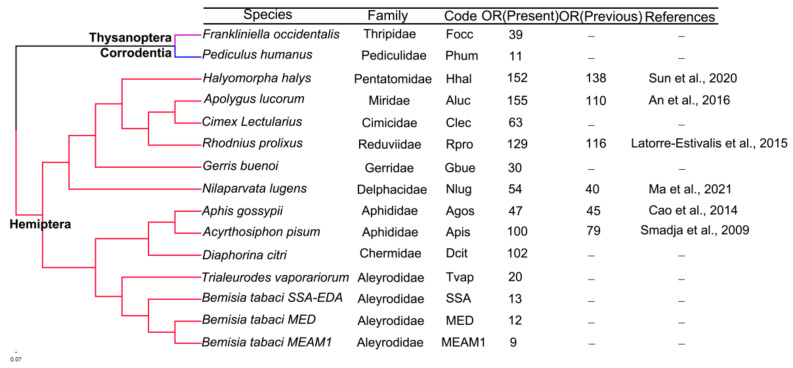
Numbers of odorant receptor genes annotated from genomes of one species of Corrodentia, one species of Thysanoptera and 10 species of Hemiptera. Phylogeny and taxonomy of the study species (left) are constructed using OrthoFinder.

**Figure 3 insects-13-00214-f003:**
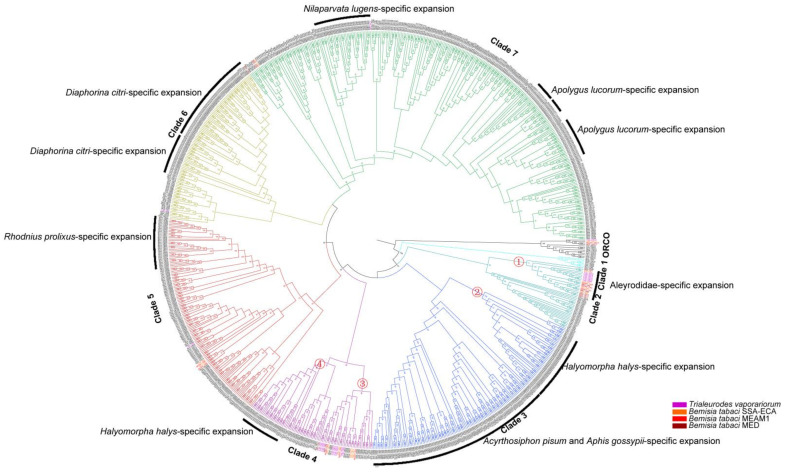
Phylogenetic relationships of the odorant receptor genes from ten species of Hemiptera: *Nilaparvata lugens* (Nlug), *Halyomorpha halys* (Hhal), *Cimex lectularius* (Clec), *Apolygus lucorum* (Aluc), *Rhodnius prolixus* (Rpro), *Gerris buenoi* (Gbue), *Diaphorina citri* (Dcit), *Acyrthosiphon pisum* (Apis), *Aphis gossypii* (Agos), *Trialeurodes vaporariorum* (Tvap), and *Bemisia tabaci* (MEAM1, MED, SSA-ECA). The phylogenetic tree was constructed from the amino acid sequence using the maximum likelihood. The tree was rooted with conserved olfactory coreceptor (Orco) genes. Nodes with support >50% were displayed in the tree.

**Figure 4 insects-13-00214-f004:**
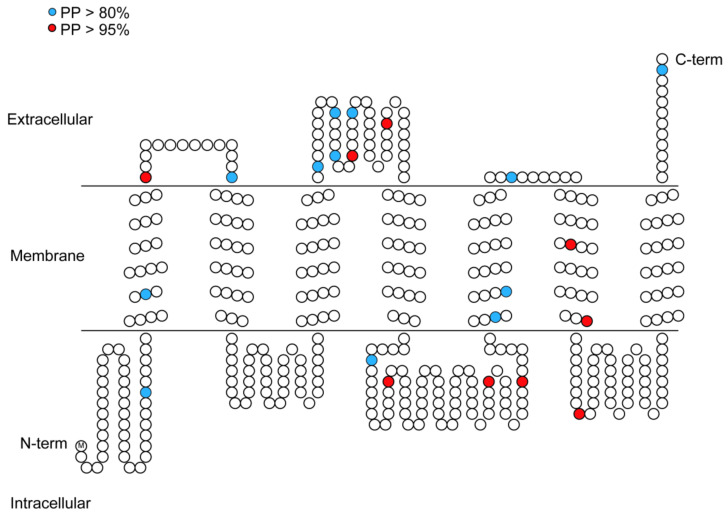
Positively selected sites on the structural topology of odorant receptors (ORs). Circles indicate the amino acids (white neutrally selected sites; blue positively selected sites with PPs 80%; red positively selected sites with PPs 95%).

**Table 1 insects-13-00214-t001:** Likelihood ratio test of the branch model for Clade 2, Clade 3, and Clade 4.

Branch	Models	LnL	2ΔL
1	M0 (one ratio)	−36,525.665006	353.227994 **
	M1 (free ratio)	−36,349.051009	
2	M0 (one ratio)	−35,609.079852	219.037414 **
	M1 (free ratio)	−35,499.561145	
3	M0 (one ratio)	−35,793.429897	420.317102 **
	M1 (free ratio)	−35,583.271346	
4	M0 (one ratio)	−78,349.502738	700.319204 **
	M1 (free ratio)	−77,999.343136	

2ΔL is twice the log-likelihood difference between models M1 and M0. ** Chi-square test indicates the difference at the high significance level of *p* < 0.01.

**Table 2 insects-13-00214-t002:** Summary of statistics for the detection of positive selection for Clade 2, Clade 3, and Clade 4 homology using the branch-site model.

Branch	LnL (nuLL)	LnL (Alternative)	2△L	P	Sites
1	−36,382.9	−36,379.1	7.4833	0.006227	NONE
2	−35,267.5	−35,258.1	18.82128	1.44 × 10^−5^	90 G 0.998 **
					206 I 0.970 *
					222 E 0.983 *
					271 V 0.954 *
					699 C 0.979 *
					711 A 0.980 *
					761 G 0.958 *
					775 G 0.989 *
					792 E 0.979 *
3	−35,664.3	−35,673.7	18.89361	1.38 × 10^−5^	NONE
4	−78,177.5	−78,168.6	17.93291	2.29 × 10^−5^	NONE

2ΔL is twice the log-likelihood difference between models M1 and M0. ** Chi-square test indicates the difference at the high significance level of *p* < 0.01. * Chi-square test indicates the difference at the significance level of *p* < 0.05.

## Data Availability

The data presented in this study are available in Appendix A.

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
