# Peer review of "Diversity and Molecular Evolution of Odorant Receptor in Hemipteran Insects"

_insects, 2022, doi:10.3390/insects13020214_

Round 1

Reviewer 1 Report

In this manuscript, Tian et. identified and analyzed 887 OR genes from 11 hemipteran species systematically. Results show that all identified Hemiptera OR genes were classified into seven major clades, and OR gene has a specific expansion in N. lugens, H. halys, A. lucorum, D. citri, A. pisum, and A. gossypii.

Some of the positive results in this manuscript might be useful for providing a reference data for molecular mechanisms of Hemiptera olfactory receptors involved in host recognition. However, the manuscript still needs to be improved in its quality as well as the writing to meet the standard of the journal. I provide my comments and suggestions for the authors as below.

  1. L56-57: Species name throughout the whole manuscript should be italizedDrosophila melanogaster”, “Tribolium castaneum” et.

  2. L96-98, Software version, reference, or web address as well as relevant parameters should be listed.

  3. L162-166,these sentences should be moved to Introduction or Discussion part.

  4. L189, it is not clear why the H. halys OR genes were selected, more explanation is needed.

  5. L239-240, somewhere, the authors need to provide an explanation for the various number of OR genes in different species.

  6. L251, this sentence is puzzling.

  7. L269, 272-273, and 276-278: How are the results relevant or related to the content of the discussion here? The authors could improve this part by providing relevant explanation to the results and Discussion.

  8. L282-293, How do these sentences relate to the results?

  9. The discussion section could be improved by emphasizing how the results from hemipteran species correspond to previous findings in other insect species.

  10. L297 (-- we identified seven major subfamilies in 11 species) and 302-303: Need more clarification or rewriting.

Author Response

Response 1: Species names have been changed to italics and checked throughout.

Response 2: The download addresses of the two databases have been added to the end of each database name.

Response 3: These sentences have been moved to the Discussion section of the paper.

Response 4: The number of ORs in Halyomorpha halys is 152, with obvious gene expansion. And the results of phylogenetic analysis showed that the OR genes of H. halys were specifically distributed in Clade 3 and Clade 4. So the OR genes of H. halys were selected.

Response 5: In Hemiptera, the biotope of different species is very different. Some species live on land, while others perched on still water, such as Gerris buenoi. In terms of feeding habits: some are herbivorous insects, some feed on blood (Rhodnius prolixus), and some feed on other small insects (Gerris buenoi). Among the herbivorous species, there are monophagous (Nilaparvata lugens), oligophagous (Laodelphax striatellus) and polyphagous (Bemisia tabaci). These factors may all cause differences in the number of ORs.

Response 6: This sentence means the number of OR genes in different species in Hemiptera have obvious gain and loss. And has been modified in the sentence.

Response 7: Similar sequences are likely to have similar spatial structures and biological functions. In Hemiptera, the functional studies of OR receptors are relatively weak. However, functional studies of some important genes have been carried out in individual species. And through the phylogenetic analysis, we are expected to speculate that some new genes in some species have similar functions to known genes, which will provide a reference for subsequent functional studies.

Response 8: Studies have shown that the acquisition of a novel host may drive the adaptive divergence of sensory systems by positive selection. In our study, we found that the OR genes of Halyomorpha halys and Bemisia tabaci were both undergoing positive selection, and positive selection amino acid sites were found in H. halys. This may explain why H. halys and B. tabaci have a wide host range.

Response 9: A comparison of the relationship between the number of ORs and hosts in Hemiptera and other insect species has been added to the Discussion. In Coleoptera, the diversity of ORs appeared to parallel the host breadth. In Hemiptera, however, there is no such correlation. This may be due to the differences in biotope and feeding habits of different species of Hemiptera. In Hemiptera, we found obvious gene gain and loss of OR number in different species and positive selection of amino acid sites, which are common in other insect orders.

Response 10: L297 means “we identified seven major subfamilies in 887 OR genes in the 11 species of Hemiptera”. And L302-303 has been rewritten as “It lays the foundation for future research into the function of ORs in hemipteran insects.”.

Reviewer 2 Report

Depending on my modest experience in this field
This manuscript aims to compare the initial sequence diversity and evolution of odorant receptors in different Hemiptera species Nilaparvata lugens, Halyomorpha halys, Cimex lectularius, Apolygus lucorum, Gerris buenoi, Rhodnius prolixus, Diaphorina citri, Acyrthosiphon pisum, Aphis gossypii, Trialeurodes vaporariorum, and Bemisia tabaci) through phylogenetic and selection pressure analysis.
However, the positive point in this research, it can help to understand the mechanism of the receptors involved in insect-plant interaction and provides important content as they identified and analyzed 887 odor receptor genes from these hemipteran species, as odorant receptors in hemipteran species are poorly investigated.
The methodology is appropriately described, and the reference is fine. However, the figures are of bad quality and difficult to read, and also the introduction is short.

Author Response

Response: The pictures in the text have all been adjusted for clarity and the size of the fonts in the picture has been adjusted. Due to the large number of OR genes in Figure 3, the details cannot be clearly presented. We have put the complete tree diagram in the appendix for readers to refer to. And the introduction has been supplemented and enriched.

Reviewer 3 Report

The authors use information from genome sequencing experiments to find and analyze odorant receptor genes in hemipterans using a bioinformatics pipeline derived from Julio Rozas, with the goal of enhancing our understanding of the evolution of this gene family.  This analysis may allow for future connections between odorant receptor structure/function and attractant/avoidance behaviors that might drive food preferences.

This is a straightforward analysis, with a strong start; however, the paper could benefit from a more thorough analysis and better depiction of the results.  For example, the text in the figures (particularly 2 and 3) are not legible, and the figures do not scale well.

For readers outside the field, it would be of benefit to know how many hemipterans have been sequenced to date.  Do the current studies reflect all available data?  If not, why were these species selected for analysis?  For readers within the field, it is nice that the authors provide sequence information for all the protein coding regions identified, but it would be nice to see additional figures or charts that summarize these sequences and features.  There is some information in the text but it comes across as an unorganized list that is rather difficult to follow.  Many features were considered:  length of sequence, exon/intron structure, etc, and this type of information should be included in a summary chart.

The ORs were defined based on sequence homology, but can one assume that is the in vivo function? The very large GPCR family encompasses proteins that do not function as odorant receptors as well.  For this reason, the text should refer to the protein-coding regions identified as “putative ORs”.  What is the likelihood that some ORs were not identified?  What is the likelihood that some ORs were misidentified?  As proof of principle, has this particular bioinformatics pipeline been used on a model organism, such as Drosophila which has a number of ORs identified empirically?  While gene expression data may be limited for these species, a discussion of the likelihood of the proteins functioning as odorant receptors would be informative.  The authors do mention transcriptome analyses for some species (line 161).

Figure 4 could use a better description of the data that led to the model. (The missing information may be within the drawing, but the text is not readable.).

Table 2 needs a better description.

In order for the paper to better withstand the “test of time”, it could use a few more references, fewer abbreviations (as some are only used once), and spelling out abbreviations in both text and figure legends.

Author Response

Response: The pictures in the text have all been adjusted for clarity and the size of the fonts in the picture has been adjusted. Due to the large number of OR genes in Figure 3, the details cannot be clearly presented. We have put the complete tree diagram in the appendix for readers to refer to.   

The species we selected in this paper cover 11 species of 9 families in the three suborders Auchenorrhyncha, Sternorrhyncha and Heteroptera. Species with different feeding habits and different habitats were included, and different biotypes of Bemisia tabaci were also selected in the Aleyrodidae. And the genomes of the selected species were all evaluated by BUSCO to ensure the quality of the genomes of the selected species. We put the sorted characteristics of the Hemipteran OR genes, such as sequence length, sequence integrity, and exon-intron structure in the appendix.

Some orthologous genes showing high sequence homology may have the similar spatial structures and biological functions. In this article, we first found potential ORs based on sequence homology. We then performed NR annotation on all of these potential OR genes, and removed those genes that were not OR. And our prediction of the transmembrane domain finally identified OR genes. Therefore, our results are a combination of multiple factors, rather than simply based on homology.

To demonstrate the availability of this bioinformatics pipeline and the advantages of annotating gene families in genomes of varying quality, previous studies have used this bioinformatics pipeline to identify chemosensory receptor genes in nine species including the model organism Drosophila melanogaster (Vizueta et al., 2017, 2018, 2020). The identification results were compared with the annotation results of augustus-ppx, and the results of this pipeline were found to be more accurate. Furthermore, to overcome putative gene model errors, the pipeline implements some filtering steps to determine whether the predicted coding sequences are correct. For example, hmmer searches were performed to identify protein family domains in all newly annotated sequences. All of these indicate that this process is reliable and feasible in the identification of insect olfactory receptor genes.

    Figure 4 and Table 2 are the results of selection pressure analysis on OR genes of Aleyrodidae and Halyomorpha halys. Among them, Table 2 is the statistical results of the positive selection of the four branches of Branch1, 2, 3, and 4 in the branch-site model. And Figure 4 is a visualization of the position of the positive selection site detected by Branch 2 in Table 2 in the OR amino acid transmembrane structure. We have added a detailed description of the selection pressure analysis analysis in the Results section to make it easier for the reader to understand.

    We have added more references to support our point, and checked all abbreviations in the article, including text and figure legends.

Round 2

Reviewer 1 Report

I am satisfied with the additions to the revised MS to address the concerns I had raised.

Author Response

We would like to thank you for your dedicated comments that helped to improve our manuscript significantly. 

Reviewer 3 Report

The revised paper is a bit confusing to review, as the "before" and "after" figures are both included, and there are some major formatting issues, especially in the discussion. 

It would be a good idea to explain the re-ordering of authors.

Figure 4 highlights positively and negatively selected sites from the analysis.  The legend could still use some clarification, as the lengths of the proteins are not identical.  For example, for the 2st positively selected site, is this amino acid always -6 from the TM aa, or is this positioning more of an estimate?

Author Response

Journal of Insects (MDPI)

Point-By-Point Explanation/Rebuttal

MS# Insects-1543081

Comments from reviewers:

The revised paper is a bit confusing to review, as the "before" and "after" figures are both included, and there are some major formatting issues, especially in the discussion. 

Response: The figures of the manuscript have been improved. We also have checked the formatting of the Discussion section and already added the necessary contents as requested.

It would be a good idea to explain the re-ordering of authors.

Response: Since Prof. Chen Luo is the Principle Investigator of the lab, at her own suggestion, she adjusted her position and put her name at the end. The order of the corresponding authors and other authors has not been modified accordingly. 

Figure 4 highlights positively and negatively selected sites from the analysis.  The legend could still use some clarification, as the lengths of the proteins are not identical. For example, for the 2st positively selected site, is this amino acid always -6 from the TM aa, or is this positioning more of an estimate?

Response: To exclude differences in sequence lengths, we first aligned the amino acid sequences of ORs of different species using MAFFT software before performing selection pressure analysis. Then, we deleted the amino acid positions where the gaps are located in the alignment results to obtain a clean data. Finally, the generated clean data is used to analyze the selection pressure of the branch site model by using the CODEML program to obtain the amino acid sites that are in positive selection in the specified branch.
